# Federated Adversarial Domain Adaptation

**Xingchao Peng**
Boston University
Boston, MA 02215, USA
xpeng@bu.edu

**Zijun Huang**
Columbia University
New York City, NY 10027, USA
zijun.huang@columbia.edu

**Yizhe Zhu**
Rutgers University
Piscataway, NJ 08854, USA
yz530@scarletmail.rutgers.edu

**Kate Saenko**
Boston University
Boston, MA 02215, USA
saenko@bu.edu

## Abstract

Federated learning improves data privacy and efficiency in machine learning performed over networks of distributed devices, such as mobile phones, IoT and wearable devices, etc. Yet models trained with federated learning can still fail to generalize to new devices due to the problem of domain shift. Domain shift occurs when the labeled data collected by source nodes statistically differs from the target node's unlabeled data. In this work, we present a principled approach to the problem of federated domain adaptation, which aims to align the representations learned among the different nodes with the data distribution of the target node. Our approach extends adversarial adaptation techniques to the constraints of the federated setting. In addition, we devise a dynamic attention mechanism and leverage feature disentanglement to enhance knowledge transfer. Empirically, we perform extensive experiments on several image and text classification tasks and show promising results under unsupervised federated domain adaptation setting.

## 1 Introduction

Data generated by networks of mobile and IoT devices poses unique challenges for training machine learning models. Due to the growing storage/computational power of these devices and concerns about data privacy, it is increasingly attractive to keep data and computation locally on the device (Smith et al., 2017). *Federated Learning (FL)* (Mohassel & Rindal, 2018; Bonawitz et al., 2017; Mohassel & Zhang, 2017) provides a privacy-preserving mechanism to leverage such decen-tralized data and computation resources to train machine learning models. The main idea behind federated learning is to have each node learn on its own local data and not share either the data or the model parameters.

While federated learning promises better privacy and efficiency, existing methods ignore the fact that the data on each node are collected in a non-*i.i.d* manner, leading to *domain shift* between nodes (Quionero-Candela et al., 2009). For example, one device may take photos mostly indoors, while another mostly outdoors. In this paper, we address the problem of transferring knowledge from the decentralized nodes to a new node with a different data domain, without requiring any additional supervision from the user. We define this novel problem *Unsupervised Federated Domain Adaptation* (UFDA), as illustrated in Figure 1(a).

There is a large body of existing work on unsupervised domain adaptation (Long et al., 2015; Ganin & Lempitsky, 2015; Tzeng et al., 2017; Zhu et al., 2017; Gong et al., 2012; Long et al., 2018), but the federated setting presents several additional challenges. First, the data are stored locally and cannot be shared, which hampers mainstream domain adaptation methods as they need to access both the labeled source and unlabeled target data (Tzeng et al., 2014; Long et al., 2017; Ghifary et al., 2016; Sun & Saenko, 2016; Ganin & Lempitsky, 2015; Tzeng et al., 2017). Second, the model parameters are trained separately for each node and converge at different speeds, while also offering different contributions to the target node depending on how close the two domains are. Finally, the knowledge

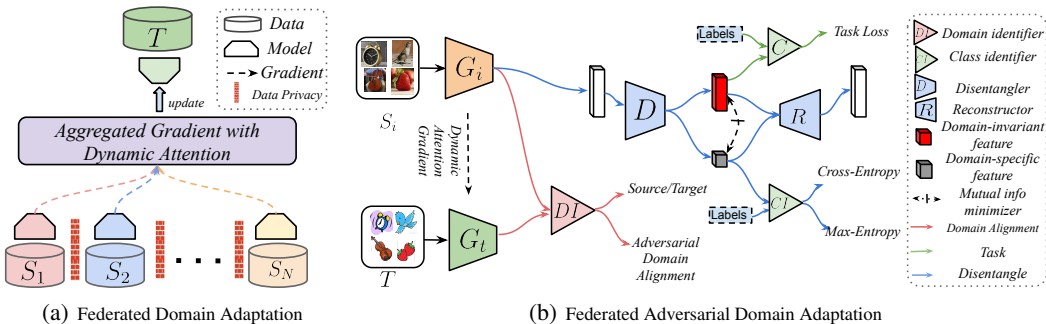

(a) Federated Domain Adaptation    (b) Federated Adversarial Domain Adaptation

Figure 1: (a) We propose an approach for the UFDA setting, where data are not shareable between different domains. In our approach, models are trained separately on each source domain and their gradients are aggregated with *dynamic attention* mechanism to update the target model. (b) Our FADA model learns to extract *domain-invariant* features using adversarial domain alignment (red lines) and a feature *disentangler* (blue lines).

learned from source nodes is highly entangled (Bengio et al., 2013), which can possibly lead to *negative transfer* (Pan & Yang, 2010).

In this paper, we propose a solution to the above problems called *Federated Adversarial Domain Adaptation* (FADA) which aims to tackle domain shift in a federated learning system through adversarial techniques. Our approach preserves data privacy by training one model per source node and updating the target model with the aggregation of source gradients, but does so in a way that reduces domain shift. First, we analyze the federated domain adaptation problem from a theoretical perspective and provide a generalization bound. Inspired by our theoretical results, we propose an efficient adaptation algorithm based on adversarial adaptation and representation disentanglement applied to the federated setting. We also devise a *dynamic attention* model to cope with the varying convergence rates in the federated learning system. We conduct extensive experiments on real-world datasets, including image recognition and natural language tasks. Compared to baseline methods, we improve adaptation performance on all tasks, demonstrating the effectiveness of our devised model.

## 2 RELATED WORK

**Unsupervised Domain Adaptation** Unsupervised Domain Adaptation (UDA) aims to transfer the knowledge learned from a labeled source domain to an unlabeled target domain. Domain adaptation approaches proposed over the past decade include discrepancy-based methods (Tzeng et al., 2014; Long et al., 2017; Ghifary et al., 2014; Sun & Saenko, 2016; Peng & Saenko, 2018), reconstruction-based UDA models (Yi et al., 2017; Zhu et al., 2017; Hoffman et al., 2018; Kim et al., 2017), and adversary-based approaches (Liu & Tuzel, 2016; Tzeng et al., 2017; Liu et al., 2018a; Ganin & Lempitsky, 2015). For example, Ganin & Lempitsky (2015) propose a *gradient reversal* layer to perform adversarial training to a domain discriminator, inspired by the idea of adversarial learning. Tzeng et al. (2017) address unsupervised domain adaptation by adapting a deep CNN-based feature extractor/classifier across source and target domains via adversarial training. Ben-David et al. (2010) introduce an $\mathcal{H}\Delta\mathcal{H}$-divergence to evaluate the domain shift and provide a generalization error bound for domain adaptation. These methods assume the data are centralized on one server, limiting their applicability to the distributed learning system.

**Federated Learning** Federated learning (Mohassel & Rindal, 2018; Rivest et al., 1978; Bonawitz et al., 2017; Mohassel & Zhang, 2017) is a decentralized learning approach which enables multiple clients to collaboratively learn a machine learning model while keeping the training data and model parameters on local devices. Inspired by Homomorphic Encryption (Rivest et al., 1978), Gilad-Bachrach et al. (2016) propose CryptoNets to enhance the efficiency of data encryption, achieving higher federated learning performance. Bonawitz et al. (2017) introduce a secure aggregation scheme to update the machine learning models under their federated learning framework. Recently, Mohassel & Zhang (2017) propose SecureML to support privacy-preserving collaborative training in a multi-client federated learning system. However, these methods mainly aim to learn a single global model across the data and have no convergence guarantee, which limits their ability to deal with non-*i.i.d.* data. To address the non-*i.i.d* data, Smith et al. (2017) introduce federated multi-task learning, which learns a separate model for each node. Liu et al. (2018b) propose semi-supervised federated transfer learning in a privacy-preserving setting. However, their models involve full or semi-supervision. The

work proposed here is, to our best knowledge, the first federated learning framework to consider unsupervised domain adaptation.

**Feature Disentanglement** Deep neural networks are known to extract features where multiple hidden factors are highly entangled. Learning disentangled representations can help remove irrelevant and domain-specific features and model only the relevant factors of data variation. To this end, recent work (Mathieu et al., 2016; Makhzani et al., 2016; Liu et al., 2018a; Odena et al., 2017) explores the learning of interpretable representations using generative adversarial networks (GANs) (Goodfellow et al., 2014) and variational autoencoders (VAEs) (Kingma & Welling, 2013). Under the fully supervised setting, (Odena et al., 2017) propose an auxiliary classifier GAN (AC-GAN) to achieve representation disentanglement. (Liu et al., 2018a) introduce a unified feature disentanglement framework to learn domain-invariant features from data across different domains. (Kingma et al., 2014) also extend VAEs into the semi-supervised setting for representation disentanglement. (Lee et al., 2018) propose to disentangle the features into a domain-invariant content space and a domain-specific attributes space, producing diverse outputs without paired training data. Inspired by these works, we propose a method to disentangle the *domain-invariant* features from *domain-specific* features, using an adversarial training process. In addition, we propose to minimize the mutual information between the *domain-invariant* features and *domain-specific* features to enhance the feature disentanglement.

## 3 GENERALIZATION BOUND FOR FEDERATED DOMAIN ADAPTATION

We first define the notation and review a typical theoretical error bound for single-source domain adaptation (Ben-David et al., 2007; Blitzer et al., 2008) devised by Ben-David *et al*. Then we describe our derived error bound for unsupervised federated domain adaptation. We mainly focus on the high-level interpretation of the error bound here and refer our readers to the appendix (see supplementary material) for proof details.

**Notation.** Let $\mathcal{D}_S$[1] and $\mathcal{D}_T$ denote source and target distribution on input space $\mathcal{X}$ and a ground-truth labeling function $g : \mathcal{X} \rightarrow \{0, 1\}$. A *hypothesis* is a function $h : \mathcal{X} \rightarrow \{0, 1\}$ with the *error* w.r.t the ground-truth labeling function $g$: $\epsilon_S(h, g) := \mathbb{E}_{\mathbf{x} \sim \mathcal{D}_S}[|h(\mathbf{x}) - g(\mathbf{x})|]$. We denote the risk and empirical risk of hypothesis $h$ on $\mathcal{D}_S$ as $\epsilon_S(h)$ and $\widehat{\epsilon}_S(h)$. Similarly, the risk and empirical risk of $h$ on $\mathcal{D}_T$ are denoted as $\epsilon_T(h)$ and $\widehat{\epsilon}_T(h)$. The $\mathcal{H}$-divergence between two distributions $\mathcal{D}$ and $\mathcal{D}'$ is defined as: $d_{\mathcal{H}}(\mathcal{D}, \mathcal{D}') := 2 \sup_{A \in \mathcal{A}_{\mathcal{H}}} |\mathrm{Pr}_{\mathcal{D}}(A) - \mathrm{Pr}_{\mathcal{D}'}(A)|$, where $\mathcal{H}$ is a hypothesis class for input space $\mathcal{X}$, and $\mathcal{A}_{\mathcal{H}}$ denotes the collection of subsets of $\mathcal{X}$ that are the support of some hypothesis in $\mathcal{H}$.

The symmetric difference space $\mathcal{H}\Delta\mathcal{H}$ is defined as: $\mathcal{H}\Delta\mathcal{H} := \{h(\mathbf{x}) \oplus h'(\mathbf{x}))|h, h' \in \mathcal{H}\}$, ($\oplus$: the XOR operation). We denote the optimal hypothesis that achieves the minimum risk on the source and the target as $h^* := \arg\min_{h \in \mathcal{H}} \epsilon_S(h) + \epsilon_T(h)$ and the error of $h^*$ as $\lambda := \epsilon_S(h^*) + \epsilon_T(h^*)$. Blitzer et al. (2007b) prove the following error bound on the target domain.

**Theorem 1.** *Let $\mathcal{H}$ be a hypothesis space of $VC$-dimension $d$ and $\widehat{\mathcal{D}}_S$, $\widehat{\mathcal{D}}_T$ be the empirical distribution induced by samples of size $m$ drawn from $\mathcal{D}_S$ and $\mathcal{D}_T$. Then with probability at least $1 - \delta$ over the choice of samples, for each $h \in \mathcal{H}$,*

$$\epsilon_T(h) \leq \widehat{\epsilon}_S(h) + \frac{1}{2}\widehat{d}_{\mathcal{H}\Delta\mathcal{H}}(\widehat{\mathcal{D}}_S, \widehat{\mathcal{D}}_T) + 4\sqrt{\frac{2d\log(2m) + \log(4/\delta)}{m}} + \lambda \tag{1}$$

Let $\mathcal{D}_S = \{\mathcal{D}_{S_i}\}_{i=1}^N$, and $\mathcal{D}_T = \{\mathbf{x}_j^t\}_{j=1}^{n_t}$ be $N$ source domains and the target domain in a UFDA system, where $\mathcal{D}_{S_i} = \{(\mathbf{x}_j^s, \mathbf{y}_j^s)\}_{j=1}^{n_i}$. In federated domain adaptation system, $\mathcal{D}_S$ is distributed on $N$ nodes and the data are not shareable with each other in the training process. The classical domain adaptation algorithms aim to minimize the target risk $\epsilon_T(h) := \mathrm{Pr}_{(\mathbf{x}, y) \sim \mathcal{D}_T}[h(\mathbf{x}) \neq y]$. However, in a UFDA system, one model cannot directly get access to data stored on different nodes for security and privacy reasons. To address this issue, we propose to learn separate models for each distributed source domain $h_S = \{h_{S_i}\}_{i=1}^N$. The target *hypothesis* $h_T$ is the aggregation of the parameters of $h_S$, *i.e.* $h_T := \sum_{i=1}^N \alpha_i h_{S_i}, \forall \alpha \in \mathbb{R}_+^N, \sum_{i \in [N]} \alpha_i = 1$. We can then derive the following error bound:

**Theorem 2.** *(Weighted error bound for federated domain adaptation). Let $\mathcal{H}$ be a hypothesis class with VC-dimension $d$ and $\{\widehat{\mathcal{D}}_{S_i}\}_{i=1}^N$, $\widehat{\mathcal{D}}_T$ be empirical distributions induced by a sample of size*

---

[1]In this literature, the calligraphic $\mathcal{D}$ denotes data distribution, and italic $D$ denotes domain discriminator.

*m from each source domain and target domain in a federated learning system, respectively. Then, $\forall \alpha \in \mathbb{R}_+^N$, $\sum_{i=1}^N \alpha_i = 1$, with probability at least $1 - \delta$ over the choice of samples, for each $h \in \mathcal{H}$,*

$$\epsilon_T(h_T) \leq \underbrace{\widehat{\epsilon}_{\tilde{S}}\Big(\sum_{i \in [N]} \alpha_i h_{S_i}\Big)}_{error\ on\ source} + \sum_{i \in [N]} \alpha_i \Big(\frac{1}{2}\underbrace{\widehat{d}_{\mathcal{H}\Delta\mathcal{H}}(\widehat{\mathcal{D}}_{S_i}, \widehat{\mathcal{D}}_T)}_{(\mathcal{D}_{S_i}, \mathcal{D}_T)\ divergence} + \lambda_i\Big) + \underbrace{4\sqrt{\frac{2d\log(2Nm) + \log(4/\delta)}{Nm}}}_{VC\text{-}Dimension\ Constraint}$$

(2)

*where $\lambda_i$ is the risk of the optimal hypothesis on the mixture of $\mathcal{D}_{S_i}$ and $T$, and $\tilde{S}$ is the mixture of source samples with size $Nm$. $\widehat{d}_{\mathcal{H}\Delta\mathcal{H}}(\widehat{\mathcal{D}}_{S_i}, \widehat{\mathcal{D}}_T)$ denotes the divergence between domain $S_i$ and $T$.*

**Comparison with Existing Bounds** The bound in (2) is extended from (1) and they are equivalent if only one source domain exists ($N = 1$). Mansour et al. (2009) provide a generalization bound for multiple-source domain adaptation, assuming that the target domain is a mixture of the $N$ source domains. In contrast, in our error bound (2), the target domain is assumed to be an novel domain, resulting in a bound involving $\mathcal{H}\Delta\mathcal{H}$ discrepancy (Ben-David et al., 2010) and the VC-dimensional constraint (Vapnik & Vapnik, 1998). Blitzer et al. (2007b) propose a generalization bound for semi-supervised multi-source domain adaptation, assuming that partial target labels are available. Our generalization bound is devised for unsupervised learning. Zhao et al. (2018) introduce classification and regression error bounds for multi-source domain adaptation. However, these error bounds assume that the multiple source and target domains exactly share the same hypothesis. In contrast, our error bound involves multiple hypotheses.

## 4 FEDERATED ADVERSARIAL DOMAIN ADAPTATION

The error bound in Theorem (2) demonstrates the importance of the weight $\alpha$ and the discrepancy $d_{\mathcal{H}\Delta\mathcal{H}}(\mathcal{D}_S, \mathcal{D}_T)$ in unsupervised federated domain adaptation. Inspired by this, we propose dynamic attention model to learn the weight $\alpha$ and federated adversarial alignment to minimize the discrepancy between the source and target domains, as shown in Figure 1. In addition, we leverage representation disentanglement to extract *domain-invariant* representations to further enhance knowledge transfer.

**Dynamic Attention** In a federated domain adaptation system, the models on different nodes have different convergence rates. In addition, the domain shifts between the source domains and target domain are different, leading to a phenomenon where some nodes may have no contribution or even *negative transfer* (Pan & Yang, 2010) to the target domain. To address this issue, we propose dynamic attention, which is a mask on the gradients from source domains. The philosophy behind the dynamic attention is to increase the weight of those nodes whose gradients are beneficial to the target domain and limit the weight of those whose gradients are detrimental to the target domain. Specifically, we leverage the *gap statistics* (Tibshirani et al., 2001) to evaluate how well the target features $f^t$ can be clustered with unsupervised clustering algorithms (K-Means). Assuming we have $k$ clusters, the *gap statistics* are computed as:

$$I = \sum_{r=1}^k \frac{1}{2n_r} \sum_{i,j \in C_r} ||f_i^t - f_j^t||_2 \tag{3}$$

where we have clusters $C_1, C_2, ..., C_k$, with $C_r$ denoting the indices of observations in cluster $r$, and $n_r = |C_r|$. Intuitively, a smaller *gap statistics* value indicates the feature distribution has smaller intra-class variance. We measure the contribution of each source domain by the *gap statistics gain* between two consecutive iterations: $I_i^{gain} = I_i^{p-1} - I_i^p$ ($p$ indicating training step), denoting how much the clusters can be improved before and after the target model is updated with the $i$-th source model's gradient. The mask on the gradients from source domains is defined as $Softmax(I_1^{gain}, I_2^{gain}, ..., I_N^{gain})$.

**Federated Adversarial Alignment** The performance of machine learning models degrades rapidly with the presence of domain discrepancy (Long et al., 2015). To address this issue, existing work (Hoffman et al., 2018; Tzeng et al., 2015) proposes to minimize the discrepancy with an adversarial training process. For example, Tzeng et al. (2015) proposes the domain confusion objective, under which the feature extractor is trained with a cross-entropy loss against a uniform distribution. However, these models require access to the source and target data simultaneously, which is prohibitive in UFDA. In the federated setting, we have multiple source domains and the data are locally stored

---

**Algorithm 1** Federated Adversarial Domain Adaptation

---

**Input:** $N$ source domains $\mathcal{D}_S=\{\mathcal{D}_{S_i}\}_{i=1}^N$; a target domain $\mathcal{D}_t = \{\mathbf{x}_j^t\}_{j=1}^{n_t}$; $N$ feature extractors $\{\Theta^{G_1}, \Theta^{G_2}, ...\Theta^{G_N}\}$, $N$ disentanglers $\{\Theta^{D_1}, \Theta^{D_2}, ...\Theta^{D_N}\}$, $N$ classifiers $\{\Theta^{C_1}, \Theta^{C_2}, ...\Theta^{C_N}\}$, $N$ class identifiers $\{\Theta^{CI_1}, \Theta^{CI_2}, ...\Theta^{CI_N}\}$, $N$ mutual information estimators $\{\Theta^{M_1}, \Theta^{M_2}, ...\Theta^{M_N}\}$ trained on source domains. Target feature extractor $\Theta^{G_t}$, classifier $\Theta^{C_t}$. $N$ domain identifiers $\{\Theta^{DI_1}, \Theta^{DI_2}, ..., \Theta^{DI_N}\}$

**Output:** well-trained target feature extractor $\hat{\Theta}^{G_t}$, target classifier $\hat{\Theta}^{C_t}$ .

Model Initialization .

1: **while** not converged **do**
2:    **for** i **do**=1:N
3:       Sample mini-batch from from $\{(\mathbf{x}_i^s, y_i^s)\}_{i=1}^{n_s}$ and $\{\mathbf{x}_j^t\}_{j=1}^{n_t}$;
4:       Compute gradient with cross-entropy classification loss, update $\Theta^{G_i}, \Theta^{C_i}$.
5:       **Domain Alignment:**
6:       Update $\Theta^{DI_i}, \{\Theta^{G_i}, \Theta^{G_t}\}$ with Eq. 4 and Eq. 5 respectively to align the domain distribution;
7:       **Domain Disentangle**:
8:       update $\Theta^{G_i}, \Theta^{D_i}, \Theta^{C_i}, \Theta^{CI_i}$ with Eq. 6
9:       update $\Theta^{D_i}$ and $\{\Theta^{G_i}\}$ with Eq. 7
10:      **Mutual Information Minimization**:
11:      Calculate mutual information between the disentangled feature pair $(f_{di}, f_{ds})$ with $M_i$;
12:      Update $\Theta^{D_i}, \Theta^{M_i}$ by Eq.8;
13:    **end for**
14:    **Dynamic weight:**
15:    Calculate dynamic weight by Eq. 3
16:    Update $\Theta^{G_t}, \Theta^{C_t}$ by aggregated $\{\Theta^{G_1}, \Theta^{G_2}, ..., \Theta^{G_N}\}$, $\{\Theta^{C_1}, \Theta^{C_2}, ...\Theta^{C_N}\}$ respectively with the computed dynamic weight;
17: **end while**
18: **return** $\Theta^{G_t}, \Theta^{C_t}$

---

in a privacy-preserving manner, which means we cannot train a single model which has access to the source domain and target domain simultaneously. To address this issue, we propose federated adversarial alignment that divides optimization into two independent steps, a domain-specific local feature extractor and a global discriminator. Specifically, (1) for each domain, we train a local feature extractor, $G_i$ for $\mathcal{D}_i$ and $G_t$ for $\mathcal{D}_t$, (2) for each $(\mathcal{D}_i, \mathcal{D}_t)$ source-target domain pair, we train an adversarial domain identifier $DI$ to align the distributions in an adversarial manner: we first train $DI$ to identify which domain are the features come from, then we train the generator $(G_i, G_t)$ to confuse the $DI$. Note that $D$ only gets access to the output vectors of $G_i$ and $G_t$, without violating the UFDA setting. Given the $i$-th source domain data $\mathbf{X}^{S_i}$, target data $\mathbf{X}^T$, the objective for $DI_i$ is defined as follows:

$$\underset{\Theta^{DI_i}}{L_{adv_{DI_i}}}(\mathbf{X}^{S_i}, \mathbf{X}^T, G_i, G_t) = -\mathbb{E}_{\mathbf{x}^{s_i} \sim \mathbf{X}^{s_i}} \left[\log DI_i(G_i(\mathbf{x}^{s_i}))\right] - \mathbb{E}_{\mathbf{x}^t \sim \mathbf{X}^t}[\log(1 - DI_i(G_t(\mathbf{x}^t)))].$$

(4)

In the second step, $\mathcal{L}_{\text{adv}_D}$ remains unchanged, but $\mathcal{L}_{\text{adv}_G}$ is updated with the following objective:

$$\underset{\Theta^{G_i}, \Theta^{G_t}}{L_{adv_G}}(\mathbf{X}^{S_i}, \mathbf{X}^T, DI_i) = -\mathbb{E}_{\mathbf{x}^{s_i} \sim \mathbf{X}^{s_i}}[\log DI_i(G_i(\mathbf{x}^{s_i}))] - \mathbb{E}_{\mathbf{x}^t \sim \mathbf{X}^t}[\log DI_i(G_t(\mathbf{x}^t))] \qquad (5)$$

**Representation Disentanglement** We employ *adversarial disentanglement* to extract the *domain-invariant* features. The high-level intuition is to disentangle the features extracted by $(G_i, G_t)$ into domain-invariant and domain-specific features. As shown in Figure 1(b), the disentangler $D_i$ separates the extracted features into two branches. Specifically, we first train the $K$-way classifier $C_i$ and $K$-way class identifier $CI_i$ to correctly predict the labels with a cross-entropy loss, based on $f_{di}$ and $f_{ds}$ features, respectively. The objective is:

$$\underset{\Theta^{G_i}, \Theta^{D_i}, \Theta^{C_i}, \Theta^{CI_i}}{L_{cross-entropy}} = -\mathbb{E}_{(\mathbf{x}^{s_i}, \mathbf{y}^{s_i}) \sim \widehat{\mathcal{D}}_{s_i}} \sum_{k=1}^K \mathbb{1}[k = \mathbf{y}^{s_i}]log(C_i(f_{di})) - \mathbb{E}_{(\mathbf{x}^{s_i}, \mathbf{y}^{s_i}) \sim \widehat{\mathcal{D}}_{s_i}} \sum_{k=1}^K \mathbb{1}[k = \mathbf{y}^{s_i}]log(CI_i(f_{ds}))$$

(6)

where $f_{di} = D_i(G_i(\mathbf{x^{s_i}}))$, $f_{ds} = D_i(G_i(\mathbf{x^{s_i}}))$ denote the *domain-invariant* and *domain-specific* features respectively. In the next step, we freeze the class identifier $CI_i$ and only train the feature disentangler to confuse the class identifier $CI_i$ by generating the *domain-specific* features $f_{ds}$, as shown in Figure 1. This can be achieved by minimizing the negative entropy loss of the predicted

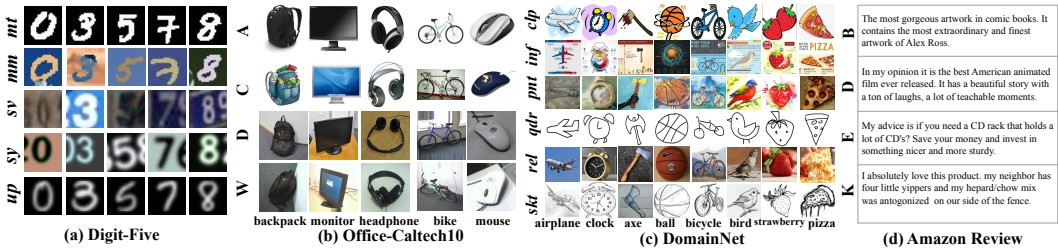

Figure 2: We demonstrate the effectiveness of FADA on four datasets: (1) "Digit-Five", which includes MNIST (*mt*), MNIST-M (*mm*), SVHN (*sv*), Synthetic (*syn*), and USPS (*up*). (2) Office-Caltech10 dataset, which contains *Amazon* (**A**), *Caltech* (**C**), *DSLR* (**D**), and *Webcam* (**W**). (3) DomainNet dataset, which includes: *clipart* (*clp*), *infograph* (*inf*), *painting* (*pnt*), *quickdraw* (*qdr*), *real* (*rel*), and *sktech* (*skt*). (4) Amazon Review dataset, which contains review for *Books* (**B**), *DVDs* (**D**), *Electronics* (**E**), and *Kitchen & housewares* (**K**).

class distribution. The objective is as follows:

$$L_{ent} \atop \Theta^{D_i},\Theta^{G_i} = -\frac{1}{N_{s_i}}\sum_{j=1}^{N_{s_i}} \log CI_i(f_{ds}^j) = -\frac{1}{N_{s_i}}\sum_{j=1}^{N_{s_i}} \log CI_i(D_i(G_i(\mathbf{x}^{s_i}))) \tag{7}$$

Feature disentanglement facilitates the knowledge transfer by reserving $f_{di}$ and dispelling $f_{ds}$. To enhance the disentanglement, we minimize the mutual information between *domain-invariant* features and *domain-specific* features, following Peng et al. (2019). Specifically, the mutual information is defined as $I(f_{di}; f_{ds}) = \int_{\mathcal{P}\times\mathcal{Q}} \log \frac{d\mathbb{P}_{\mathcal{P}\mathcal{Q}}}{d\mathbb{P}_{\mathcal{P}}\otimes\mathbb{P}_{\mathcal{Q}}} d\mathbb{P}_{\mathcal{P}\mathcal{Q}}$, where $\mathbb{P}_{\mathcal{P}\mathcal{Q}}$ is the joint probability distribution of $(f_{di}, f_{ds})$, and $\mathbb{P}_{\mathcal{P}} = \int_{\mathcal{Q}} d\mathbb{P}_{\mathcal{P}\mathcal{Q}}$, $\mathbb{P}_{\mathcal{Q}} = \int_{\mathcal{P}} d\mathbb{P}_{\mathcal{P}\mathcal{Q}}$ are the marginals. Despite being a pivotal measure across different distributions, the mutual information is only tractable for discrete variables, for a limited family of problems where the probability distributions are unknown (Belghazi et al., 2018). Following Peng et al. (2019), we adopt the Mutual Information Neural Estimator (MINE) (Belghazi et al., 2018) to estimate the mutual information by leveraging a neural network $T_\theta$: $\widehat{I(\mathcal{P};\mathcal{Q})}_n = \sup_{\theta\in\Theta} \mathbb{E}_{\mathbb{P}_{\mathcal{P}\mathcal{Q}}^{(n)}}[T_\theta] - \log(\mathbb{E}_{\mathbb{P}_{\mathcal{P}}^{(n)}\otimes\widehat{\mathbb{P}}_{\mathcal{Q}}^{(n)}}[e^{T_\theta}])$. Practically, MINE can be calculated as $I(\mathcal{P};\mathcal{Q}) = \int\int \mathbb{P}_{\mathcal{P}\mathcal{Q}}^n(p,q) T(p,q,\theta) - \log(\int\int \mathbb{P}_{\mathcal{P}}^n(p)\mathbb{P}_{\mathcal{Q}}^n(q)e^{T(p,q,\theta)})$. To avoid computing the integrals, we leverage Monte-Carlo integration to calculate the estimation:

$$I(\mathcal{P}, \mathcal{Q}) = \frac{1}{n}\sum_{i=1}^n T(p,q,\theta) - \log(\frac{1}{n}\sum_{i=1}^n e^{T(p,q',\theta)}) \tag{8}$$

where $(p, q)$ are sampled from the joint distribution, $q'$ is sampled from the marginal distribution, and $T(p, q, \theta)$ is the neural network parameterized by $\theta$ to estimate the mutual information between $\mathcal{P}$ and $\mathcal{Q}$, we refer the reader to MINE (Belghazi et al., 2018) for more details. The *domain-invariant* and *domain-specific* features are forwarded to a reconstructor with a L2 loss to reconstruct the original features, aming to keep the representation integrity, as shown in Figure 1(b).

**Optimization** Our model is trained in an end-to-end fashion. We train federated alignment and representation disentanglement component with Stochastic Gradient Descent (Kiefer et al., 1952). The federated adversarial alignment loss and representation disentanglement loss are minimized together with the task loss. The detailed training procedure is presented in Algorithm 1.

## 5 EXPERIMENTS

We test our model on the following tasks: digit classification (*Digit-Five*), object recognition (*Office-Caltech10* (Gong et al., 2012), *DomainNet* (Peng et al., 2018)) and sentiment analysis (*Amazon Review* dataset (Blitzer et al., 2007a)). Figure 2 shows some data samples and Table 9 (see supplementary material) shows the number of data per domain we used in our experiments. We perform our experiments on a 10 Titan-Xp GPU cluster and simulate the federated system on a single machine (as the data communication is not the main focus of this paper). Our model is implemented with PyTorch. We repeat every experiment 10 times on the *Digit-Five* and *Amazon Review* datasets, and 5 times on the *Office-Caltech10* and *DomainNet* (Peng et al., 2018) datasets, reporting the mean and standard derivation of accuracy. To better explore the effectiveness of different components of our model, we propose three different ablations, *i.e.* model **I**: with *dynamic attention*; model **II**: **I** + *adversarial alignment*; and model **III**: **II** + *representation disentanglement*.

| Models | mt,sv,sy,up→mm | mm,sv,sy,up→mt | mt,mm,sy,up→sv | mt,mm,sv,up→sy | mt,mm,sv,sy→up | Avg |
|---|---|---|---|---|---|---|
| Source Only | 63.3±0.7 | 90.5±0.8 | 88.7±0.8 | 63.5±0.9 | 82.4±0.6 | 77.7 |
| DAN | 63.7±0.7 | 96.3±0.5 | 94.2±0.8 | 62.4±0.7 | 85.4±0.7 | 80.4 |
| DANN | 71.3±0.5 | 97.6±0.7 | 92.3±0.8 | 63.4±0.7 | 85.3±0.8 | 82.1 |
| Source Only | 49.6±0.8 | 75.4±1.3 | 22.7±0.9 | 44.3±0.7 | 75.5±1.4 | 53.5 |
| AdaBN | 59.3±0.8 | 75.3±0.7 | 34.2±0.6 | 59.7±0.7 | 87.1±0.9 | 61.3 |
| AutoDIAL | 60.7±1.6 | 76.8±0.9 | 32.4±0.5 | 58.7±1.2 | 90.3±0.9 | 65.8 |
| *f*-DANN | 59.5±0.6 | 86.1±1.1 | 44.3 ±0.6 | 53.4±0.9 | 89.7±0.9 | 66.6 |
| *f*-DAN | 57.5 ±0.8 | 86.4 ±0.7 | 45.3±0.7 | 58.4±0.7 | 90.8 ±1.1 | 67.7 |
| **FADA**+*attention* (**I**) | 44.2±0.7 | 90.5±0.8 | 27.8±0.5 | 55.6±0.8 | 88.3±1.2 | 61.3 |
| **FADA**+*adversarial* (**II**) | 58.2±0.8 | **92.5**± 0.9 | 48.3±0.6 | 62.1±0.5 | 90.6±1.1 | 70.3 |
| **FADA**+*disentangle* (**III**) | **62.5**±0.7 | 91.4 ±0.7 | **50.5** ±0.3 | **71.8**±0.5 | **91.7**±1.0 | **73.6** |

Table 1: Accuracy (%) on "Digit-Five" dataset with UFDA protocol. FADA achieves **73.6**%, outperforming other baselines. We incrementally add each component t our model, aiming to study their effectiveness on the final results. (model **I**: with *dynamic attention*; model **II**: **I**+*adversarial alignment*; model **III**: **II**+*representation disentanglement*. *mt*, *up*, *sv*, *sy*, *mm* are abbreviations for *MNIST*, *USPS*, *SVHN*, *Synthetic Digits*, *MNIST-M*.)

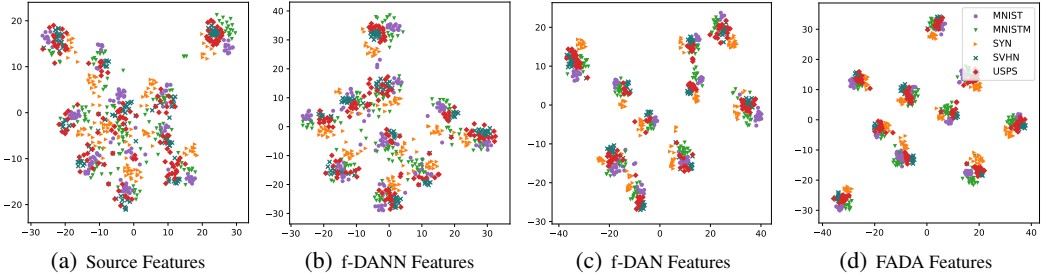

(a) Source Features  (b) f-DANN Features  (c) f-DAN Features  (d) FADA Features

Figure 3: Feature visualization: t-SNE plot of source-only features, f-DANN (Ganin & Lempitsky, 2015) features, f-DAN (Long et al., 2015) features and FADA features in *sv,mm,mt,sy→up* setting. We use different markers and colors to denote different domains. The data points from target domain have been denoted by red for better visual effect. (Best viewed in color.)

## 5.1 EXPERIMENTS ON DIGIT RECOGNITION

**Digit-Five** This dataset is a collection of five benchmarks for digit recognition, namely *MNIST* (LeCun et al., 1998), *Synthetic Digits* (Ganin & Lempitsky, 2015), *MNIST-M* (Ganin & Lempitsky, 2015), *SVHN*, and *USPS*. In our experiments, we take turns setting one domain as the target domain and the rest as the distributed source domains, leading to five transfer tasks. The detailed architecture of our model can be found in Table 7 (see supplementary material).

Since many DA models (Saito et al., 2018; French et al., 2018; Hoffman et al., 2018) require access to data from different domains, it is infeasible to directly compare our model to these baselines. Instead, we compare our model to the following popular domain adaptation baselines: Domain Adversarial Neural Network (**DANN**) (Ganin & Lempitsky, 2015), Deep Adaptation Network (**DAN**) (Long et al., 2015), Automatic DomaIn Alignment Layers (**AutoDIAL**) (Carlucci et al., 2017), and Adaptive Batch Normalization (**AdaBN**) Li et al. (2016). Specifically, DANN minimizes the domain gap between source domain and target domain with a *gradient reversal* layer. DAN applies multi-kernel MMD loss (Gretton et al., 2007) to align the source domain with the target domain in Reproducing Kernel Hilbert Space. AutoDIAL introduces domain alignment layer to deep models to match the source and target feature distributions to a reference one. AdaBN applies Batch Normalization layer (Ioffe & Szegedy, 2015) to facilitate the knowledge transfer between the source and target domains. When conducting the baseline experiments, we use the code provided by the authors and modify the original settings to fit federated DA setting (*i.e.* each domain has its own model), denoted by *f*-DAN and *f*-DANN. In addition, to demonstrate the difficulty of UFDA where accessing all source data with a single model is prohibitive, we also perform the corresponding *multi-source DA* experiments (shared source data).

**Results and Analysis** The experimental results are shown in Table 1. From the results, we can make the following observations. (1) Model **III** achieves 73.6% average accuracy, significantly

| Method | C,D,W → A | A,D,W → C | A,C,W → D | A,C,D → W | Average |
|---|---|---|---|---|---|
| AlexNet | 80.1±0.4 | 86.9±0.3 | 82.7±0.5 | 85.1±0.3 | 83.7 |
| $f$-DAN | 82.5±0.5 | 87.2±0.4 | 85.6±0.4 | 86.1±0.3 | 85.4 |
| $f$-DANN | 83.1±0.4 | 86.5±0.5 | 84.8±0.5 | 86.4±0.5 | 85.2 |
| **FADA**+*attention* (**I**) | 81.2±0.3 | 87.1±0.6 | 83.5±0.5 | 85.9±0.4 | 84.4 |
| **FADA**+*adversarial* (**II**) | 83.1±0.6 | 87.8±0.4 | 85.4±0.4 | 86.8±0.5 | 85.8 |
| **FADA**+*disentangle* (**III**) | **84.3**±0.6 | 88.4±0.5 | 86.1±0.4 | 87.3±0.5 | 86.5 |
| ResNet101 | 81.9±0.5 | 87.9±0.3 | 85.7±0.5 | 86.9±0.4 | 85.6 |
| AdaBN | 82.2±0.4 | 88.2±0.6 | 85.9±0.7 | 87.4±0.8 | 85.7 |
| AutoDIAL | 83.3±0.6 | 87.7±0.8 | 85.6±0.7 | 87.1±0.6 | 85.9 |
| $f$-DAN | 82.7±0.3 | 88.1±0.5 | 86.5±0.3 | 86.5±0.3 | 85.9 |
| $f$-DANN | 83.5±0.4 | 88.5±0.3 | 85.9±0.5 | 87.1±0.4 | 86.3 |
| **FADA**+*attention* (**I**) | 82.1±0.5 | 87.5±0.3 | 85.8±0.4 | 87.3±0.5 | 85.7 |
| **FADA**+*adversarial* (**II**) | 83.2±0.4 | 88.4±0.3 | 86.4±0.5 | 87.8±0.4 | 86.5 |
| **FADA**+*disentangle* (**III**) | 84.2±0.5 | **88.7**±0.5 | **87.1**±0.6 | **88.1**±0.4 | **87.1** |

Table 2: Accuracy on *Office-Caltech10* dataset with unsupervised federated domain adaptation protocol. The upper table shows the results for AlexNet backbone and the table below shows the results for ResNet backbone.

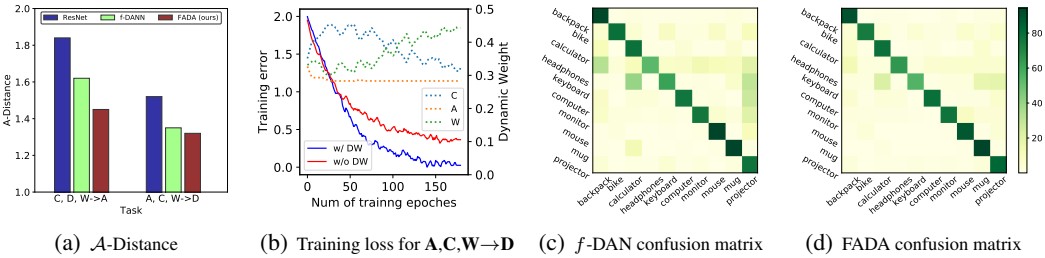

(a) $\mathcal{A}$-Distance  (b) Training loss for **A,C,W→D**  (c) $f$-DAN confusion matrix  (d) FADA confusion matrix

Figure 4: (**a**)$\mathcal{A}$-Distance of ResNet, $f$-DANN, and FADA features on two different tasks. (**b**) training errors and dynamic weight on **A,C,W→D** task. (**c**)-(**d**) confusion matrices of $f$-DAN, and FADA on **A,C,D→W** task.

outperforming the baselines. (2) The results of model **I** and model **II** demonstrate the effectiveness of *dynamic attention* and *adversarial alignment*. (3) Federated DA displays much weaker results than *multi-source DA*, demonstrating that the newly proposed UFDA learning setting is very challenging.

To dive deeper into the feature representation of our model versus other baselines, we plot in Figure 3(a)-3(d) the t-SNE embeddings of the feature representations learned on *mm,mt,sv,sy→up* task with source-only features, $f$-DANN features, $f$-DAN features and FADA features, respectively. We observe that the feature embeddings of our model have smaller intra-class variance and larger inter-class variance than $f$-DANN and $f$-DAN, demonstrating that our model is capable of generating the desired feature embedding and can extract *domain-invariant* features across different domains.

## 5.2 EXPERIMENTS ON OFFICE-CALTECH10

**Office-Caltech10** (Gong et al., 2012) This dataset contains 10 common categories shared by Office-31 (Saenko et al., 2010) and Caltech-256 datasets (Griffin et al., 2007). It contains four domains: *Caltech* (**C**), which are sampled from Caltech-256 dataset, *Amazon* (**A**), which contains images collected from amazon.com, *Webcam* (**W**) and *DSLR* (**D**), which contains images taken by web camera and DSLR camera under office environment.

We leverage two popular networks as the backbone of feature generator $G$, *i.e.* AlexNet (Krizhevsky et al., 2012) and ResNet (He et al., 2016). Both the networks are pre-trained on ImageNet (Deng et al., 2009). Other components of our model are randomly initialized with the normal distribution. In the learning process, we set the learning rate of randomly initialized parameters to ten times of the pre-trained parameters as it will take more time for those parameters to converge. Details of our model are listed in Table 9 (supplementary material).

| Models | inf,pnt,qdr, rel,skt→clp | clp,pnt,qdr, rel,skt→inf | clp,inf,qdr, rel,skt→pnt | clp,inf,pnt, rel,skt→qdr | clp,inf,pnt, qdr,skt→rel | clp,inf,pnt, qdr,rel→skt | Avg |
|---|---|---|---|---|---|---|---|
| AlexNet | 39.2±0.7 | 12.7±0.4 | 32.7±0.4 | 5.9±0.7 | 40.3±0.5 | 22.7±0.6 | 25.6 |
| $f$-DAN | 41.6±0.6 | 13.7±0.5 | 36.3±0.5 | 6.5±0.5 | 43.5±0.8 | 22.9±0.5 | 27.4 |
| $f$-DANN | 42.6±0.8 | 14.1±0.7 | 35.2±0.3 | 6.2±0.7 | 42.9±0.5 | 22.7±0.7 | 27.2 |
| **FADA**+*disentangle* (**III**) | 44.9±0.7 | 15.9±0.6 | 36.3±0.8 | **8.6**±0.8 | 44.5±0.6 | 23.2±0.8 | 28.9 |
| ResNet101 | 41.6 ±0.6 | 14.5±0.7 | 35.7±0.7 | 8.4±0.7 | 43.5±0.7 | 23.3±0.7 | 27.7 |
| $f$-DAN | 43.5±0.7 | 14.1±0.6 | 37.6±0.7 | 8.3±0.6 | 44.5±0.5 | 25.1±0.5 | 28.9 |
| $f$-DANN | 43.1±0.8 | 15.2±0.9 | 35.7±0.4 | 8.2±0.6 | 45.2±0.7 | **27.1**±0.6 | 29.1 |
| **FADA**+*disentangle* (**III**) | **45.3**±0.7 | **16.3**±0.8 | **38.9** ±0.7 | 7.9±0.4 | **46.7**±0.4 | 26.8±0.4 | **30.3** |

Table 3: Accuracy (%) on the DomainNet dataset (Peng et al., 2018) dataset under UFDA protocol. The upper table shows the results based on AlexNet (Krizhevsky et al., 2012) backbone and the table below are the results based on ResNet (He et al., 2016) backbone.

| Method | D,E,K → B | B,E,K → D | B,D,K → E | B,D,E → K | Average |
|---|---|---|---|---|---|
| Source Only | 74.4±0.3 | 79.2±0.4 | 73.5 ±0.2 | 71.4±0.1 | 74.6 |
| $f$-DANN | 75.2±0.3 | **82.7**±0.2 | 76.5±0.3 | 72.8±0.4 | 76.8 |
| AdaBN | 76.7±0.3 | 80.9±0.3 | 75.7±0.2 | 74.6±0.3 | 76.9 |
| AutoDIAL | 76.3±0.4 | 81.3±0.5 | 74.8±0.4 | 75.6±0.2 | 77.1 |
| $f$-DAN | 75.6±0.2 | 81.6±0.3 | **77.9**±0.1 | 73.2±0.2 | 77.6 |
| **FADA**+*attention* (**I**) | 74.8±0.2 | 78.9±0.2 | 74.5±0.3 | 72.5±0.2 | 75.2 |
| **FADA**+*adversarial* (**II**) | **79.7**±0.2 | 81.1±0.1 | 77.3±0.2 | 76.4±0.2 | 78.6 |
| **FADA**+*disentangle* (**III**) | 78.1±0.2 | **82.7**±0.1 | 77.4±0.2 | **77.5**±0.3 | **78.9** |

Table 4: Accuracy (%) on "Amazon Review" dataset with unsupervised federated domain adaptation protocol.

**Results and Analysis** The experimental results on Office-Caltech10 datasets are shown in Table 2. We utilize the same backbones as the baselines and separately show the results. We make the following observations from the results: **(1)** Our model achieves **86.5**% accuracy with an AlexNet backbone and **87.1**% accuracy with a ResNet backbone, outperforming the compared baselines. **(2)** All the models have similar performance when **C**,**D**,**W** are selected as the target domain, but perform worse when **A** is selected as the target domain. This phenomenon is probably caused by the large domain gap, as the images in **A** are collected from amazon.com and contain a white background.

To better analyze the effectiveness of FADA, we perform the following empirical analysis: **(1)** $\mathcal{A}$-**distance** Ben-David et al. (2010) suggests $\mathcal{A}$-distance as a measure of domain discrepancy. Following Long et al. (2015), we calculate the approximate $\mathcal{A}$-distance $\hat{d}_{\mathcal{A}} = 2\,(1 - 2\epsilon)$ for **C**,**D**,**W**→**A** and **A**,**C**,**W**→**D** tasks, where $\epsilon$ is the generalization error of a two-sample classifier (e.g. kernel SVM) trained on the binary problem of distinguishing input samples between the source and target domains. In Figure 4(a), we plot for tasks with raw ResNet features, $f$-DANN features, and FADA features, respectively. We observe that the $\hat{d}_{\mathcal{A}}$ on DADA features are smaller than ResNet features and $f$-DANN features, demonstrating that FADA features are harder to be distinguished between source and target. **(2)** To show how the dynamic attention mechanism benefits the training process, we plot the training loss *w/* or *w/o* dynamic weights for **A**,**C**,**W**→**D** task in Figure 4(b). The figure shows the target model's training error is much smaller when dynamic attention is applied, which is consistent with the quantitative results. In addition, in **A**,**C**,**W**→**D** setting, the weight of **A** decreases to the lower bound after first a few epochs and the weight of **W** increases during the training process, as photos in both **D** and **W** are taken in the same environment with different cameras. **(3)** To better analyze the error mode, we plot the confusion matrices for $f$-DAN and FADA on **A**,**C**,**D**->**W** task in Figure 4(c)-4(d). The figures show that $f$-DAN mainly confuses "calculator" *vs.* "keyboard", "backpack" with "headphones", while FADA is able to distinguish them with disentangled features.

## 5.3 EXPERIMENTS ON DOMAINNET

**DomainNet** [2] This dataset contains approximately 0.6 million images distributed among 345 categories. It comprises of six domains: *Clipart* (*clp*), a collection of clipart images; *Infograph* (*inf*),

---

[2] http://ai.bu.edu/M3SDA/

| target | mm | mt | sv | sy | up | Avg | A | C | D | W | Avg | B | D | E | K | Avg |
|---|---|---|---|---|---|---|---|---|---|---|---|---|---|---|---|---|
| **FADA** *w/o. attention* | 60.1 | 91.2 | 49.2 | 69.1 | 90.2 | **71.9** | 83.3 | 85.7 | 86.2 | 88.3 | **85.8** | 77.2 | 82.8 | 77.2 | 76.3 | **78.3** |
| **FADA** *w. attention* | 62.5 | 91.4 | 50.5 | 71.8 | 91.7 | **73.6** | 84.2 | 88.7 | 87.1 | 88.1 | **87.1** | 78.1 | 82.7 | 77.4 | 77.5 | **78.9** |

Table 5: The ablation study results show that the dynamic attention module is essential for our model.

infographic images with specific object; *Painting* (*pnt*), artistic depictions of object in the form of paintings; *Quickdraw* (*qdr*), drawings from the worldwide players of game "Quick Draw!"[3]; *Real* (*rel*, photos and real world images; and *Sketch* (*skt*), sketches of specific objects. This dataset is very large-scale and contains rich and informative vision cues across different domains, providing a good testbed for unsupervised federated domain adaptation. Some sample images can be found in Figure 2.

**Results** The experimental results on DomainNet are shown in Table 3. Our model achieves **28.9**% and **30.3**% accuracy with AlexNet and ResNet backbone, respectively. In both scenarios, our model outperforms the baselines, demonstrating the effectiveness of our model on large-scale dataset. Note that this dataset contains about 0.6 million images, and so even a one-percent performance improvement is not trivial. From the experiment results, we can observe that all the models deliver less desirable performance when *infograph* and *quickdraw* are selected as the target domains. This phenomenon is mainly caused by the large domain shift between *inf/qdr* domain and other domains.

### 5.4 EXPERIMENTS ON AMAZON REVIEW

**Amazon Review** (Blitzer et al., 2007a) This dataset provides a testbed for cross-domain sentimental analysis of text. The task is to identify whether the sentiment of the reviews is positive or negative. The dataset contains reviews from `amazon.com` users for four popular merchandise categories: *Books* (**B**), *DVDs* (**D**), *Electronics* (**E**), and *Kitchen appliances* (**K**). Following Gong et al. (2013), we utilize 400-dimensional bag-of-words representation and leverage a fully connected deep neural network as the backbone. The detailed architecture of our model can be found in Table 8 (supplementary materials).

**Results** The experimental results on Amazon Review dataset are shown in Table 4. Our model achieves an accuracy of **78.9**% and outperforms the compared baselines. We make two major observations from the results: (1) Our model is not only effective on vision tasks but also performs well on linguistic tasks under UFDA learning schema. (2) From the results of model **I** and **II**, we can observe the dynamic attention and federated adversarial alignment are beneficial to improve the performance. However, the performance boost from Model **II** to Model **III** is limited. This phenomenon shows that the linguistic features are harder to disentangle comparing to visual features.

### 5.5 ABLATION STUDY

To demonstrate the effectiveness of dynamic attention, we perform the ablation study analysis. The Table 5 shows the results on "Digit-Five", Office-Caltech10 and Amazon Review benchmark. We observe that the performance drops in most of the experiments when dynamic attention model is **not** applied. The dynamic attention model is devised to cope with ***the varying convergence rates in the federated learning system***, *i.e.*, different source domains have their own convergence rate. In addition, it will increase the weight of a specific domain when the domain shift between that domain and the target domain is small, and decrease the weight otherwise.

## 6 CONCLUSION

In this paper, we first proposed a novel unsupervised federated domain adaptation (UFDA) problem and derived a theoretical generalization bound for UFDA. Inspired by the theoretical results, we proposed a novel model called Federated Adversarial Domain Adaptation (FADA) to transfer the knowledge learned from distributed source domains to an unlabeled target domain with a novel dynamic attention schema. Empirically, we showed that feature disentanglement boosts the performance of FADA in UFDA tasks. An extensive empirical evaluation on UFDA vision and linguistic benchmarks demonstrated the efficacy of FADA against several domain adaptation baselines.

---

[3]`https://quickdraw.withgoogle.com/data`

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

## 7 NOTATIONS

We provide the explanations of notations occurred in this paper.

| Notations | Name | Details |
|---|---|---|
| $DI$ | domain identifier | align the domain pair $(G_i, G_t)$ |
| $D$ | disentangler | disentangle the feature to $f_{di}$ and feature $f_{ds}$. |
| $C$ | classifier | predict the labels for input features |
| $CI$ | class identify | extract domain-specific features. |
| $G_i$ | feature generator | extract features for $\mathcal{D}_{S_i}$ |
| $G_t$ | feature generator | extract features for $\mathcal{D}_t$ |
| $M$ | mutual information estimator | estimate the mutual information between $f_{ds}$ and $f_{di}$ |

Table 6: Notations occurred in the paper.

## 8 MODEL ARCHITECTURE

We provide the detailed model architecture (Table 7 and Table 9) for each component in our model: Generator, Disentangler, Domain Classifier, Classifier and MINE.

| layer | configuration |
|---|---|
| | Feature Generator |
| 1 | Conv2D (3, 64, 5, 1, 2), BN, ReLU, MaxPool |
| 2 | Conv2D (64, 64, 5, 1, 2), BN, ReLU, MaxPool |
| 3 | Conv2D (64, 128, 5, 1, 2), BN, ReLU |
| | Disentangler |
| 1 | FC (8192, 3072), BN, ReLU |
| 2 | DropOut (0.5), FC (3072, 2048), BN, ReLU |
| | Domain Identifier |
| 1 | FC (2048, 256), LeakyReLU |
| 2 | FC (256, 2), LeakyReLU |
| | Class Identifier |
| 1 | FC (2048, 10), BN, Softmax |
| | Reconstructor |
| 1 | FC (4096, 8192) |
| | Mutual Information Estimator |
| fc1_x | FC (2048, 512), LeakyReLU |
| fc1_y | FC (2048, 512), LeakyReLU |
| 2 | FC (512,1) |

Table 7: Model architecture for digit recognition task ("Digit-Five" dataset). For each convolution layer, we list the input dimension, output dimension, kernel size, stride, and padding. For the fully-connected layer, we provide the input and output dimensions. For drop-out layers, we provide the probability of an element to be zeroed.

## 9 DETAILS OF DATASETS

We provide the detailed information of datasets. For Digit-Five and DomainNet, we provide the train/test split for each domain. For Office-Caltech10, we provide the number of images in each domain. For Amazon review dataset, we show the detailed number of positive reviews and negative reviews for each merchandise category.

| layer | configuration |
|---|---|
| Feature Generator | |
| 1 | FC (400, 128), BN, ReLU |
| Disentangler | |
| 1 | FC (128, 64), BN, ReLU |
| 2 | DropOut (0.5), FC (64, 32), BN, ReLU |
| Domain Identifier | |
| 1 | FC (64, 32), LeakyReLU |
| 2 | FC (32, 2), LeakyReLU |
| Class Identifier | |
| 1 | FC (32, 2), BN, Softmax |
| Reconstructor | |
| 1 | FC (64, 128) |
| Mutual Information Estimator | |
| fc1_x | FC (32, 16), LeakyReLU |
| fc1_y | FC (32, 16), LeakyReLU |
| 2 | FC (16,1) |

Table 8: Model architecture for cross-doman sentimental analysis task ("Amazon Review" dataset (Blitzer et al., 2007a)). For the fully-connected layers (FC), we provide the input and output dimensions. For drop-out layers (Dropout), we provide the probability of an element to be zeroed.

| layer | configuration |
|---|---|
| Feature Generator: ResNet101 or AlexNet | |
| Disentangler | |
| 1 | Dropout(0.5), FC (2048, 2048), BN, ReLU |
| 2 | Dropout(0.5), FC (2048, 2048), BN, ReLU |
| Domain Identifier | |
| 1 | FC (2048, 256), LeakyReLU |
| 2 | FC (256, 2), LeakyReLU |
| Class Identifier | |
| 1 | FC (2048, 10), BN, Softmax |
| Reconstructor | |
| 1 | FC (4096, 2048) |
| Mutual Information Estimator | |
| fc1_x | FC (2048, 512), LeakyReLU |
| fc1_y | FC (2048, 512), LeakyReLU |
| 2 | FC (512,1) |

Table 9: Model architecture for image recognition task (Office-Caltech10 (Gong et al., 2012) and DomainNet (Peng et al., 2018)). For each convolution layer, we list the input dimension, output dimension, kernel size, stride, and padding. For the fully-connected layer, we provide the input and output dimensions. For drop-out layers, we provide the probability of an element to be zeroed.

| Digit-Five | | | | | | | |
|---|---|---|---|---|---|---|---|
| Splits | *mnist* | *mnist_m* | *svhn* | *syn* | *usps* | | Total |
| Train | 25,000 | 25,000 | 25,000 | 25,000 | 7,348 | | 107,348 |
| Test | 9,000 | 9,000 | 9,000 | 9,000 | 1,860 | | 37,860 |
| Office-Caltech10 | | | | | | | |
| Splits | | *Amazon* | *Caltech* | *Dslr* | *Webcam* | | Total |
| Total | | 958 | 1,123 | 157 | 295 | | 2,533 |
| DomainNet | | | | | | | |
| Splits | *clp* | *inf* | *pnt* | *qdr* | *rel* | *skt* | Total |
| Train | 34,019 | 37,087 | 52,867 | 120,750 | 122,563 | 49,115 | 416,401 |
| Test | 14,818 | 16,114 | 22,892 | 51,750 | 52,764 | 21,271 | 179,609 |
| Amazon Review | | | | | | | |
| Splits | | *Books* | *DVDs* | *Electronics* | *Kitchen* | | Total |
| Positive | | 1,000 | 1,000 | 1,000 | 1,000 | | 4,000 |
| Negative | | 1,000 | 1,000 | 1,000 | 1,000 | | 4,000 |

Table 10: Detailed number of samples we used in our experiments.

## 10 PROOF OF THEOREM 2

**Theorem 3.** *(Weighted error bound for federated domain adaptation). Let $\mathcal{H}$ be a hypothesis class with VC-dimension d and $\{\widehat{\mathcal{D}}_{S_i}\}_{i=1}^N$, $\widehat{\mathcal{D}}_T$ be empirical distributions induced by a sample of size m from each source domain and target domain in a federated learning system, respectively. Then, $\forall \alpha \in \mathbb{R}_+^N$, $\sum_{i=1}^N \alpha_i = 1$, with probability at least $1 - \delta$ over the choice of samples, for each $h \in \mathcal{H}$,*

$$\epsilon_T(h_T) \leq \underbrace{\widehat{\epsilon}_{\tilde{S}}\Big(\sum_{i\in[N]} \alpha_i h_{S_i}\Big)}_{error\ on\ source} + \sum_{i\in[N]} \alpha_i\Big(\frac{1}{2}\underbrace{\widehat{d}_{\mathcal{H}\Delta\mathcal{H}}(\widehat{\mathcal{D}}_{S_i},\widehat{\mathcal{D}}_T)}_{(\mathcal{D}_{S_i},\mathcal{D}_T)\ divergence}+\lambda_i\Big) + \underbrace{4\sqrt{\frac{2d\log(2Nm)+\log(4/\delta)}{Nm}}}_{VC\text{-}Dimension\ Constraint}$$

(9)

*where $\lambda_i$ is the risk of the optimal hypothesis on the mixture of $\mathcal{D}_{S_i}$ and $T$, and $\tilde{S}$ is the mixture of source samples with size $Nm$.*

*Proof.* Consider a combined source domain which is equivalent to a mixture distribution of the $N$ source domains, with the mixture weight $\alpha$, where $\alpha \in \mathbb{R}_+^N$ and $\sum_{i=1}^N \alpha_i = 1$. Denote the mixture source domain distribution as $\tilde{D}_S^\alpha$ (where $\tilde{D}_S^\alpha := \sum_{i\in[N]} \alpha_i \mathcal{D}_{S_i}$), and the data sampled from $\tilde{D}_S^\alpha$ as $\tilde{S}$. Theoretically, we can assume $\tilde{D}_S^\alpha$ and $\mathcal{D}_T$ to be the source domain and target domain, respectively. Apply Theorem 1, we have that for $0 < \delta < 1$, with probability of at least 1-$\delta$ over the choice of samples, for each $h \in \mathcal{H}$,

$$\epsilon_T(h) \leq \widehat{\epsilon}_{\tilde{S}}(h) + \frac{1}{2}\widehat{d}_{\mathcal{H}\Delta\mathcal{H}}(\widehat{\mathcal{D}}_{\tilde{S}},\widehat{\mathcal{D}}_T) + 4\sqrt{\frac{2d\log(2Nm)+\log(4/\delta)}{Nm}} + \lambda_\alpha \qquad (10)$$

where $\lambda_\alpha$ is the risk of optimal hypothesis on the $\tilde{S}$ and $T$. The upper bound of $\widehat{d}_{\mathcal{H}\Delta\mathcal{H}}(\widehat{\mathcal{D}}_{\tilde{S}},\widehat{\mathcal{D}}_T)$ can be derived as follows:

$$\begin{aligned}
\widehat{d}_{\mathcal{H}\Delta\mathcal{H}}(\widehat{\mathcal{D}}_{\tilde{S}},\widehat{\mathcal{D}}_T) &= 2 \sup_{A\in\mathcal{A}_{\mathcal{H}\Delta\mathcal{H}}} |\Pr_{\widehat{\mathcal{D}}_{\tilde{S}}}(A) - \Pr_{\widehat{\mathcal{D}}_T}(A)| \\
&= 2 \sup_{A\in\mathcal{A}_{\mathcal{H}\Delta\mathcal{H}}} |\sum_{i\in[N]} \alpha_i(\Pr_{\widehat{\mathcal{D}}_{\tilde{S}_i}}(A) - \Pr_{\widehat{\mathcal{D}}_T}(A))| \\
&\leqslant 2 \sup_{A\in\mathcal{A}_{\mathcal{H}\Delta\mathcal{H}}} \sum_{i\in[N]} \alpha_i(|\Pr_{\widehat{\mathcal{D}}_{\tilde{S}_i}}(A) - \Pr_{\widehat{\mathcal{D}}_T}(A))| \\
&\leqslant 2 \sum_{i\in[N]} \alpha_i \sup_{A\in\mathcal{A}_{\mathcal{H}\Delta\mathcal{H}}} (|\Pr_{\widehat{\mathcal{D}}_{\tilde{S}_i}}(A) - \Pr_{\widehat{\mathcal{D}}_T}(A))| \\
&= \sum_{i\in[N]} \alpha_i\widehat{d}_{\mathcal{H}\Delta\mathcal{H}}(\widehat{\mathcal{D}}_{S_i},\widehat{D}_T)
\end{aligned}$$

the first inequality is derived by the triangle inequality. Similarly, with the triangle inequality property, we can derive $\lambda_\alpha \leqslant \sum_{i\in[N]} \alpha_i\lambda_i$. On the other hand, for $\forall h_T \in \mathcal{H}$, we have: $\widehat{\epsilon}_{\tilde{S}}(h_T) = \widehat{\epsilon}_{\tilde{S}}(\sum_{i\in[N]} \alpha_i h_{S_i})$. Replace $\widehat{\epsilon}_{\tilde{S}}(h)$, $\lambda_\alpha$ and $\widehat{d}_{\mathcal{H}\Delta\mathcal{H}}(\widehat{\mathcal{D}}_{\tilde{S}},\widehat{\mathcal{D}}_T)$ in Eq. 10, we have:

$$\begin{aligned}
\epsilon_T(h_T) &\leq \widehat{\epsilon}_{\tilde{S}}(h_T) + \frac{1}{2}\widehat{d}_{\mathcal{H}\Delta\mathcal{H}}(\widehat{\mathcal{D}}_{\tilde{S}},\widehat{\mathcal{D}}_T) + 4\sqrt{\frac{2d\log(2Nm)+\log(4/\delta)}{Nm}} + \lambda_\alpha \\
&= \widehat{\epsilon}_{\tilde{S}}\Big(\sum_{i\in[N]} \alpha_i h_{S_i}\Big) + \frac{1}{2}\widehat{d}_{\mathcal{H}\Delta\mathcal{H}}(\widehat{\mathcal{D}}_{\tilde{S}},\widehat{\mathcal{D}}_T) + 4\sqrt{\frac{2d\log(2Nm)+\log(4/\delta)}{Nm}} + \lambda_\alpha \\
&\leq \widehat{\epsilon}_{\tilde{S}}\Big(\sum_{i\in[N]} \alpha_i h_{S_i}\Big) + \frac{1}{2}\sum_{i\in[N]} \alpha_i\widehat{d}_{\mathcal{H}\Delta\mathcal{H}}(\widehat{\mathcal{D}}_{S_i},\widehat{D}_T) + \sum_{i\in[N]} \alpha_i\lambda_i + 4\sqrt{\frac{2d\log(2Nm)+\log(4/\delta)}{Nm}} \\
&= \underbrace{\widehat{\epsilon}_{\tilde{S}}\Big(\sum_{i\in[N]} \alpha_i h_{S_i}\Big)}_{error\ on\ source} + \sum_{i\in[N]} \alpha_i\Big(\frac{1}{2}\underbrace{\widehat{d}_{\mathcal{H}\Delta\mathcal{H}}(\widehat{\mathcal{D}}_{S_i},\widehat{\mathcal{D}}_T)}_{(\mathcal{D}_{S_i},\mathcal{D}_T)\ divergence}+\lambda_i\Big) + \underbrace{4\sqrt{\frac{2d\log(2Nm)+\log(4/\delta)}{Nm}}}_{VC\text{-}Dimension\ Constraint}
\end{aligned}$$

$\square$

**Remark.** The equation in Theorem 2 provides a theoretical error bound for unsupervised federated domain adaptation as it assumes that the source data distributed on different nodes can form a mixture source domain. In fact, the data on different node can not be shared under the federated learning schema. The theoretical error bound is only valid when the weights of models on all the nodes are fully synchronized.

