# OpenReview forum: "Federated Adversarial Domain Adaptation"
_ICLR.cc/2020/Conference — Accept (Poster)_

### Official Review · AnonReviewer2 · 2019-10-23
**Official Blind Review #2**

**Rating:** 6

**Review:**

The paper proposed and studied the unsupervised federated domain adaption problem, which aims to transfer knowledge from source nodes to a new node with different data distribution. To address the problem, a federated adversarial domain adaption (FADA) algorithm is introduced in the paper. The key idea of the algorithm is to update the target model by aggregating the gradients from source nodes, and also leverage adversarial adaption techniques to reduce the discrepancy between source features and target features. Overall, the problem studied in the paper is interesting, theoretical analysis on the error bound is provided in the paper, and the effectiveness of the proposed method has been validated in various datasets. Although the technical contributions of the paper are solid, I still have several concerns about it.
1. The proposed algorithm is not described very clearly in section 4. According to the paper, DI is used to identify the domain from the output of Gi and Gt and align the features from those domains, then how is it related to the disentanglement in Eq 6. Also in Eq 6, symbol C_s was not introduced in the previous context, which makes it confusing to understand this objective.

2. It would be better if the author(s) can provide some complexity analysis of the proposed algorithm.

3. The paper still contains some typos and unresolved reference issues.

**Experience Assessment:**

I have read many papers in this area.

**Review Assessment: Checking Correctness Of Derivations And Theory:**

I assessed the sensibility of the derivations and theory.

**Review Assessment: Checking Correctness Of Experiments:**

I assessed the sensibility of the experiments.

**Review Assessment: Thoroughness In Paper Reading:**

I read the paper at least twice and used my best judgement in assessing the paper.

---

> ### Author Response · Authors · 2019-11-13
> **Algorithm in Sec 4 clarified, complexity analysis given, typos and unresolved reference issues fixed**
>
> We thank the reviewer for the overall positive feedback on our paper. We have updated the details of the algorithm in section 4.
>
> 1. The proposed algorithm is not described very clearly in Section 4.
>
> Thank you for pointing this out.  We admit that the notations in the Federated Adversarial Alignment and Representation Disentanglement part are unclear, which has also been mentioned by Reviewer #1. We want to mention the following response is the same with the response to Reviewer #1, 2nd concerns.
>
> We have revised the paper as follows to make it more clear (refer to the revision paper):
> i) we standardized the notation occurred in the paper and made a table to explain each notation in the supplementary material (Table 6).
> ii) we update Figure1(b) and Algorithm 1 to make the notations consistent and clear.
> iii) we edit Equation 4~7 to clarify which parameters will the loss optimizes on.
> iv) we have added high-level intuition to our paper to help readers better understand the method section.
>
> -In summary, we clarify the notation in the Federated Adversarial Alignment and Representation Disentanglement as follows:
>
> $DI$        |   domain identifier    | align the domain pair ($G_i$,$G_t$)
> $D$          |   disentangler             | disentangle the feature to domain-invariant feature $f_{di}$ and domain-specific feature $f_{ds}$.
> $C$          |   classifier                    | predict the labels for input features
> $CI$         |   class identify           | first trained with task loss, then train with the adversarial loss to extract domain-specific features.
> $G_i$      |feature generator       | extract features for $\mathcal{D}_{S_i}$
> $G_t$      | feature generator      | extract features for ${\mathcal{D}}_t$
> $M$          | MI estimator               | estimate the mutual information between $f_{di}$ and $f_{ds}$.
>
> -Equation 4 has been revised to:
> $$\underset{\Theta^{DI_i}}{\mathcal{L}_{{adv}_{DI_i}}}(\mathbf{X}^{S_i}, \mathbf{X}^T, G_i, G_t)=- \mathbb{E}_{\mathbf{x}^{s_i} \sim \mathbf{X}^{s_i}}\left[\log DI_i(G_i(\mathbf{x}^{s_i}))]-\mathbb{E}_{\mathbf{x}^t \sim \mathbf{X}^t} [\log(1 - DI_i(G_t(\mathbf{x}^{t})))\right]$$
>
> The $\mathcal{L}_{{adv}_{DI}}$ will update the parameter of $DI$, i.e. $\Theta^{DI}$.
>
> -Equation 5 has been revised to:
> $$\underset{\Theta^{G_i},\Theta^{G_t}}{\mathcal{L}_{{adv}_G}} (\mathbf{X}^{S_i}, \mathbf{X}^T, DI_i) = -
>     \mathbb{E}_{\mathbf{x}^{s_i} \sim \mathbf{X}^{s_i}}[\log DI_i(G_{i}(\mathbf{x}^{s_i}))]-
>     \mathbb{E}_{\mathbf{x}^t \sim \mathbf{X}^t}[\log DI_i(G_t(\mathbf{x}^t))]$$
>
> The $\mathcal{L}_{{adv}_G}$ will update the parameter of $G_i$ and $G_t$, i.e. $\Theta^{G_i}$ and $\Theta^{G_t}$.
>
> These two objectives are to train the domain classifier $DI$  and ($G_i$, $G_t$) in an adversarial manner: we first train $DI$ to identify which domain are the features come from, then we train the generator ($G_i$, $G_t$) to confuse the $DI$. The aim is to align the distributions of the features generated by ($G_i$, $G_t$).
>
> -Equation 6 has been revised to:
> $$\underset{\Theta^{G_i},\Theta^{D_i},\Theta^{C_i},\Theta^{CI_i}}{\mathcal{L}_{cross-entropy}} = -\mathbb{E}_{(\mathbf{x}^{s_i},\mathbf{y}^{s_i})\sim\widehat{\mathcal{D}}_{s_i}} \sum_{k=1}^{K}1 [k=\mathbf{y}^{s_i}]log(C_i(f_{di}))
>     -\mathbb{E}_{(\mathbf{x}^{s_i},\mathbf{y}^{s_i})\sim\widehat{\mathcal{D}}_{s_i}} \sum_{k=1}^{K}1 [k=\mathbf{y}^{s_i}]log(CI_i(f_{ds}))$$
>
> The $\mathcal{L}_{cross-entropy}$ is minimized over the parameters of the feature generator $G_i$, the disentangler $D_i$, the classifier $C$, and the class identifier $CI$, i.e. $\Theta^{G_i}$, $\Theta^{D_i}$, $\Theta^{C}$, $\Theta^{CI}$.
>
> -Equation 7 has been revised to:
> $$\underset{\Theta^{D_i},\Theta^{G_i}}{\mathcal{L}_{ent}} = - \frac{1}{N_{s_i}} \sum_{j=1}^{N_{s_i}} \log CI_i(f^j_{ds}) = - \frac{1}{N_{s_i}} \sum_{j=1}^{N_{s_i}} \log CI_i(D_i(G_i(\mathbf{x}^{s_i})))$$
>
> The ${\mathcal{L}_{ent}}$ is minimized over the parameters of the feature generator $G_i$, the disentangler $D_i$, i.e. $\Theta^{G_i}$, $\Theta^{D_i}$.
> Feature disentanglement facilitates the knowledge transfer by extracting  $f_{di}$ and dispelling $f_{ds}$
>
> 2. Complexity analysis of the proposed algorithm.
>
> -From Algorithm 1, we observe that the four processes, i.e. feature extraction, domain alignment, domain disentangle, and mutual information minimization are linear with N, where N is the number of source domains. Thus, the time complexity of operations on deep neural networks is O(N).
>
> -The most time-consuming part of the dynamic attention model is the KMeans algorithm. The average complexity is given by $O(knT)$, where $k$ is the number of clusters, $n$ is the number of samples from the target domain, and $T$ is the number of iteration for the KMeans algorithm. Take digit experiment as an example, k=10, n=128, T=1000, it takes 0.21 second.
>
> 3. Typos and unresolved reference issues
>
> We have revised the typos unresolved reference in the updated pdf submission.

---

### Official Review · AnonReviewer1 · 2019-10-24
**Official Blind Review #1**

**Rating:** 3

**Review:**

The authors present a novel algorithm for dealing with domain adaptation in the setting of federated learning (classification, specifically). That is, they tackle the issue of learning a model on a new domain when access to the data points used in training the source models is not possible due to privacy constraints. The approach uses the gradients of the source models, reweighed to account for the differing shifts between the different sources and the target domain, to fit the model on the target domain.

The authors motivate their approach by providing a novel bound on the generalization error of transfer learning when the hypothesis function used on the target domain is a convex combination of hypotheses fitted on multiple source domains. This bound shows that the weighted sum of divergences in the symmetric difference hypothesis space controls the generalization error, so the authors aim at deriving feature representations and using aggregation weights that ensure this weighted sum is small.

The authors use a novel dynamic attention model to get the aggregation weights: they cluster the features in the target domain, and measure how much the intra-cluster variation decreases when information from a given source domain is incorporated. The aggregation weights for the model updates on the target domain are then weighed using a softmax transform of these contribution weights.

The motivation up through and including section 3 is clear, the theoretical results are presented clearly, but the model details in section 4 are unclear:
-  In the dynamic attention mechanism, how does one a priori choose the number of clusters in computing the gap statistics, and what is the impact?
- the notation in the federated adversarial alignment section is unclear: what *exactly* are the model coefficients Theta that are being updated?
- the statement "optimize following objective" is made several times. this is ambiguous, and should be corrected to "miminize" following objective.
- the representation disentanglement process is intricate, and only vaguely addressed. how does one fit the neural net and use (8)? where is the l2 reconstruction loss balanced with the mutual information? the vagueness of this section means Algorithm 1 is not well-specified.

The experiments are reasonable, and compare to baseline domain adaptation methods.

The problem considered is of interest, and the approach is novel and interesting. However, the algorithm is not described in sufficient detail. After reading the paper, and spending considerable time rereading section 4, I still do not understand how Algorithm 1 is implemented in practice. For that reason I lean towards reject. I will update my score if the authors clarify the details of Algorithm 1.

Comments:
- the symmetric difference hypothesis space is incorrectly called the HdeltaH divergence in section 3


**Experience Assessment:**

I do not know much about this area.

**Review Assessment: Checking Correctness Of Derivations And Theory:**

I assessed the sensibility of the derivations and theory.

**Review Assessment: Checking Correctness Of Experiments:**

I assessed the sensibility of the experiments.

**Review Assessment: Thoroughness In Paper Reading:**

I read the paper thoroughly.

---

> ### Author Response · Authors · 2019-11-14
> **[rebuttal 2/2]  Typos addresses, Details of Algorithm 1 addressed, Implementations of our model given, Balance of l2 loss and mutual information given**
>
> [Rebuttal Part 2/2]
>
>
> The ${\mathcal{L}_{ent}}$ is minimized over the parameters of the feature generator $G_i$, the disentangler $D_i$, i.e. $\Theta^{G_i}$, $\Theta^{D_i}$.
> Feature disentanglement facilitates the knowledge transfer by extracting  $f_{di}$ and dispelling $f_{ds}$
>
>
> 3. "Optimize following objective" should be "minimizing following objective".
>
> We thank the reviewer for pointing this out. We have revised it in the revision paper.
>
> 4. The representation disentanglement process is intricate and only vaguely addressed. How to fit the neural net and use (8)? Where is l2 reconstruction loss balanced with the mutual information?
>
> -We have revised the  representation disentanglement section (refer to the red text in our paper)
>
> -To fit our model to the neural network, one should implement each module in our paper and connect them following Figure 1(b). We have released the hyper-parameters of each module in the supplemental materials. To better illustrate this, we have revised all the equations in our paper with the detailed module names. (Also, we have released our code in the submission).
>
> -To use Equation (8), one should implement the $T(p,q, \Theta )$ as a neural network. We adopt the implementation of the paper "Mutual Information Neural Estimator" in our code.
>
> -The balance of the l2 reconstruction and mutual information can be achieved by adjusting the hyper-parameters of the l2 loss and mutual information loss. We revised the paper to better illustrate this.
>
> 5.  Comments: symmetric difference space is incorrectly called $\mathcal{H}\Delta\mathcal{H}$.
>
> We have revised this typo in the revision paper. Thanks again for pointing this out!

---

> ### Author Response · Authors · 2019-11-14
> **[Rebuttal 1/2]  Algorithm 1 Clarified, Paper Revised, Notations Clarified**
>
> [Rebuttal 1/2]
>
> We thank the reviewer for the positive feedback and constructive suggestions on our paper. We have revised our paper accordingly  (refer to the red text in our revised paper).
>
> 1. In the dynamic attention mechanism, how to choose the number of clusters in computing the gap statistics and what is the impact?
>
> -Since we study the close-set domain adaptation, the number of categories in the target domain and the source domain is the same. So the number of clusters is set to be the same with the number of categories in the source domain. This is the optimal and most intuitive way to set the cluster numbers, though it has some limitations in the real application since we cannot assume that the target data contains at least one example for each category.
>
> -Empirically, we study the effect of the number of clusters on "Digit-Five" dataset (Table 1 in the paper).
> --------------------------------------------------------------------------
> #cluster            |     7     |    8    |      9    |     10     |     11    |      12     |     13     |
> result (acc.)      |  72.4  | 73.1  | 73.5    |   73.6    |    73.6  |     72.8   |   72.9    |
> --------------------------------------------------------------------------
>
> 2. The notation in the federated adversarial alignment section is unclear: what exactly are the model coefficients $\Theta$ are being updated?
>
> -We admit that the notations in the Federated Adversarial Alignment and Representation Disentanglement part is unclear. We have revised the paper as follows to make it more clear (refer to the revision paper):
> We have revised the paper as follows to make it more clear (refer to the revision paper):
> i) we standardized the notation occurred in the paper and made a table to explain each notation in the supplementary material.
> ii) we update Figure1(b) and Algorithm 1 to make the notations consistent and clear.
> iii) we edit Equation 4~7 to clarify which parameters will the loss optimizes on.
> iv) we have added high-level intuition to our paper to help readers better understand the method section.
>
> -In summary, we clarify the notation in the Federated Adversarial Alignment and Representation Disentanglement as follows:
>
> $DI$        |   domain identifier    | align the domain pair ($G_i$,$G_t$)
> $D$          |   disentangler             | disentangle the feature to domain-invariant feature $f_{di}$ and domain-specific feature $f_{ds}$.
> $C$          |   classifier                    | predict the labels for input features
> $CI$         |   class identify           | first trained with task loss, then train with the adversarial loss to extract domain-specific features.
> $G_i$      |feature generator       | extract features for $\mathcal{D}_{S_i}$
> $G_t$      | feature generator      | extract features for ${\mathcal{D}}_t$
> $M$          | MI estimator               | estimate the mutual information between $f_{di}$ and $f_{ds}$.
>
> -Equation 4 has been revised to:
> $$\underset{\Theta^{DI_i}}{\mathcal{L}_{{adv}_{DI_i}}}(\mathbf{X}^{S_i}, \mathbf{X}^T, G_i, G_t)=- \mathbb{E}_{\mathbf{x}^{s_i} \sim \mathbf{X}^{s_i}}\left[\log DI_i(G_i(\mathbf{x}^{s_i}))]-\mathbb{E}_{\mathbf{x}^t \sim \mathbf{X}^t} [\log(1 - DI_i(G_t(\mathbf{x}^{t})))\right]$$
>
> The $\mathcal{L}_{{adv}_{DI}}$ will update the parameter of $DI$, i.e. $\Theta^{DI}$.
>
> -Equation 5 has been revised to:
> $$\underset{\Theta^{G_i},\Theta^{G_t}}{\mathcal{L}_{{adv}_G}} (\mathbf{X}^{S_i}, \mathbf{X}^T, DI_i) = -
>     \mathbb{E}_{\mathbf{x}^{s_i} \sim \mathbf{X}^{s_i}}[\log DI_i(G_{i}(\mathbf{x}^{s_i}))]-
>     \mathbb{E}_{\mathbf{x}^t \sim \mathbf{X}^t}[\log DI_i(G_t(\mathbf{x}^t))]$$
>
> The $\mathcal{L}_{{adv}_G}$ will update the parameter of $G_i$ and $G_t$, i.e. $\Theta^{G_i}$ and $\Theta^{G_t}$.
>
> These two objectives are to train the domain classifier $DI$  and ($G_i$, $G_t$) in an adversarial manner: we first train $DI$ to identify which domain are the features come from, then we train the generator ($G_i$, $G_t$) to confuse the $DI$. The aim is to align the distributions of the features generated by ($G_i$, $G_t$).
>
> -Equation 6 has been revised to:
> $$\underset{\Theta^{G_i},\Theta^{D_i},\Theta^{C_i},\Theta^{CI_i}}{\mathcal{L}_{cross-entropy}} = -\mathbb{E}_{(\mathbf{x}^{s_i},\mathbf{y}^{s_i})\sim\widehat{\mathcal{D}}_{s_i}} \sum_{k=1}^{K}{1} [k=\mathbf{y}^{s_i}]log(C_i(f_{di}))
>     -\mathbb{E}_{(\mathbf{x}^{s_i},\mathbf{y}^{s_i})\sim\widehat{\mathcal{D}}_{s_i}} \sum_{k=1}^{K}{1} [k=\mathbf{y}^{s_i}]log(CI_i(f_{ds}))$$
>
> The $\mathcal{L}_{cross-entropy}$ is minimized over the parameters of the feature generator $G_i$, the disentangler $D_i$, the classifier $C$, and the class identifier $CI$, i.e. $\Theta^{G_i}$, $\Theta^{D_i}$, $\Theta^{C}$, $\Theta^{CI}$.
>
> -Equation 7 has been revised to:
> $$\underset{\Theta^{D_i},\Theta^{G_i}}{\mathcal{L}_{ent}} = - \frac{1}{N_{s_i}} \sum_{j=1}^{N_{s_i}} \log CI_i(f^j_{ds}) = - \frac{1}{N_{s_i}} \sum_{j=1}^{N_{s_i}} \log CI_i(D_i(G_i(\mathbf{x}^{s_i})))$$

---

### Official Review · AnonReviewer3 · 2019-10-26
**Official Blind Review #3**

**Rating:** 6

**Review:**

This paper introduces an unsupervised federated domain adaptation (UFDA) problem and proposes a new model called Federated Adversarial Domain Adaptation (FADA) to transfer the knowledge learned from distributed source domains to an unlabeled target domain. This paper uses a dynamic attention mechanism by leveraging the gap statistics to transfer distributed source knowledge. This paper also proposes a method to disentangle the domain-invariant features from domain-specific features, using adversarial training. Moreover, a theoretical generalization bound for UFDA is derived. An extensive empirical evaluation is performed on UFDA vision and linguistic benchmarks.

This paper should be rejected because the total pipeline seems ad-hoc except for optimizing the weight of the source domain in the attention mechanism. Although the derivation of generalization bound for FDA in Sec.3 is excellent, it only demonstrates the importance of the weight $\alpha$. This result is trivial if we assume to have the same source domain as the target and utterly unrelated source domain to the target domain. It seems that proving why minimizing the gap statistics contributes to FADA is more essential in the dynamic attention mechanism. Because representation disentanglement has no relation with the derived theory, it would be better to clarify whether this method is original or not.

In the UFDA setting, the reviewer has doubts about whether it is realistic that the source node has a rich labeled data assuming our smartphones. Also, the assumption that the system cannot access the source data but must access all source feature seems a significant limitation in terms of privacy issues and communication cost between the target node and the source nodes.

It is unclear what is the final target classifier. If the target can access the teaching signal (e.g., labels or tags) in the source domains, it would be better to mention whether this situation violates the assumption the authors raised or not.

Minor comments
1) What is T(p, q, \theta) in the section of Representation Disentanglement?

2) What is C_s in eq.6? C_{s_i}?

3) In Fig.3, it is not proper to discuss the size of intra-class variance by just looking at the figures because the t-SNE is a non-linear mapping. It is better to show quantitative scores, such as the value of the Fisher criterion.

**Experience Assessment:**

I have published in this field for several years.

**Review Assessment: Checking Correctness Of Derivations And Theory:**

I assessed the sensibility of the derivations and theory.

**Review Assessment: Checking Correctness Of Experiments:**

I assessed the sensibility of the experiments.

**Review Assessment: Thoroughness In Paper Reading:**

I read the paper at least twice and used my best judgement in assessing the paper.

---

> ### Author Response · Authors · 2019-11-14
> **[Rebuttal 2/2] Misunderstanding Clarified; Minor Comments Addressed**
>
> [Rebuttal 2/2]
>
> 5. Whether it is realistic that the source node has a rich labeled data assuming the smartphone.
>
> -We are not assuming that our framework works narrowly on the smartphone network. In fact, our model works on data generated by networks of mobile, IoT devices, or other networks in which data privacy is important and domain shift is significant.
>
> -Federated learning aims to accumulate the gradient from different nodes to train a large model. In our case, we are aiming to train a good target model based on the gradient from multiple source domains. When the source node has no rich labeled data, our model can accumulate the gradient from each source domain and train a good model.
>
> 6. The assumption that the system must access all source features seems a significant limitation in terms of privacy issue and communication cost.
>
> -As shown in Figure 1(a), the model and the data are locally stored. The global discriminator only gets access to the output vectors. Since the model of each domain is not shared, it is infeasible to reconstruct the data for a specific domain using the output vectors of that domain. Even though the feature space is aligned, it is still impractical to recover the original data without the parameters of the feature extractor.
>
> -In terms of the communication cost, the source domains need to send a small-size feature vector for each data (4kb for 1024-dimensional  float feature vectors). In addition, the communication cost for transmitting the feature vector is smaller than that for transmitting the gradient of neural networks. Transmitting the gradient of neural networks is a basic setting for federated learning [6].
>
> 7. It is unclear what is the final target classifier.
>
> -The target domain is updated with the gradients from classifiers trained on the source domain since there is no label information in the target domain. (We have updated the Algorithm 1 and adversarial feature alignment section to make this clear. Refer to the red text in the revision paper)
>
> 8. If the target can access the teaching signal (e.g. labels or tags) in the source domain, it would be better to mention whether this situation violates the assumption the authors raised or not.
>
> -The target cannot get access to labels or tags from the source domain. The target domain can get access to the gradients from the source domain without violating the federated learning assumption.
>
> Minor Comments
>
> 1. What is $T(p,q,\theta)$?
>
> $T(p,q,\theta)$ is the neural network parameteralized by $\theta$ to estimate the mutual information between $\mathcal{P}$ and $\mathcal{Q}$, we refer the reviewer to "mutual information nueral network" [7] for more details.
>
> 2. What is $C_s$ in Eq.6?
>
> The $C_s$ is the class identifier, which helps to extract the domain-specific features. We have clarified and revised the notations in our revision paper. We also updated Figure 1 and Algorithm 1 to better illustrate our approach.
>
>
> [1] Federated Machine Learning: Concept and Applications. Qiang Yang et al. ACM Trans. Intell. Syst. Technol. 2019
> [2] Moment Matching for Multi-Source Domain Adaptation. Peng et al. ICCV 2019.
> [3] Towards a Definition of Disentangled Representations. Higgins, et al. DeepMind 2018.
> [4] Estimating the number of clusters in a dataset via the gap statistic. Journal of the Royal Statistical Society, 2001.
> [5] Representation learning: A review and new perspectives. Bengio Yoshua, PAMI 2013
> [6] Federated Learning: Strategies for Improving Communication Efficiency. Jakub Konečný, NIPS Workshop 2016.
> [7 ] MINE: Mutual Information Neural Estimation. Mohamed Ishmael Belghazi, ICML 2018.

---

> ### Author Response · Authors · 2019-11-14
> **[Rebuttal 1/2] Misunderstandings clarified; Why minimizing gap statistics help FADA proved;**
>
> [Rebuttal 1/2]
>
> We appreciate the reviewer for the constructive feedback on our paper!
>
> 1. The total pipeline is ad-hoc
>
> -Our pipeline is not ad-hoc designed as we are aiming to generally solve the issues caused by the domain shift in the federated learning system. We want to mention that the data heterogeneity between different nodes is one of the main limitations of the federated artificial intelligence system (Yang et al [1]). Our approach extends adversarial adaptation techniques to the constraints of the federated setting and provides a generalized, applicable and robust solution to tackle the intrinsic domain shift existing in the federated learning system.
>
> -The techniques proposed in our paper are also heuristic for conventional domain adaptation without the federated setting.
> i). Since multiple nodes exist in the federated learning (FL) system, how to efficiently leverage the useful information distributed in multiple nodes becomes a critical research topic. We propose a dynamic attention mechanism to evaluate the importance of each source domain to the target domain. This idea can be generalized to traditional multi-source domain adaptation [2].
> ii). The idea of learning domain-invariant knowledge by feature disentanglement can be generalized to conventional domain adaptation.   The disentangled representation learning in our framework posits that the FL system will benefit from separating out the underlying structure of the deep features [3].
>
> 2. Although the derivation of generalization bound for FDA in Sec.3 is excellent, it only demonstrates the importance of the weight.
>
> -We would like to claim that the error bound in Theorem (2) of Sec.3 demonstrates the importance of the weight $\alpha$, as well as the discrepancy ${d}_{\mathcal{H}\Delta\mathcal{H}}({\mathcal{D}}_{S},{\mathcal{D}}_T)$.
>
> Inspired by this, we develop the following mechanism towards tackling UFDA task: i) since the weight $\alpha$ is important for the upper bound, we propose a dynamic attention model to learn the weight $\alpha$ based on the gap statistics [4]; ii) as the discrepancy  ${d}_{\mathcal{H}\Delta\mathcal{H}}({\mathcal{D}}_{S},{\mathcal{D}}_T)$ is a critical factor for the error bound, we utilize federated adversarial alignment to minimize the discrepancy between the source and target domains.
>
>  -The error bound implies that the generalization error can be bounded by the optimal source error and the domain discrepancy.
>
> 3. Proving why minimizing the gap statistics of K-Means contributes to FADA is more essential in the dynamic attention mechanism.
>
> -We prove that minimizing the gap statistics contributes to FADA by showing that the optimal $\alpha$ can be derived by the gap statistics loss between two consecutive iterations.
> i) A smaller gap statistics value indicates the feature distribution has smaller intra-class variance.
> ii) A better classifier can be trained if the intra-class variance is smaller.
> iii) When the gradients from specific nodes are beneficial to decreasing the gap statistics, we increase the $\alpha$ corresponding to these nodes, otherwise, we decrease the $\alpha$.
> iv) Once the optimal $\alpha$ is derived by minimizing the gap statistics, we have can train a better classifier on the target domain.
>
> 4. Representation disentanglement has no relation with the derived theory, it would be better to clarify whether this method is original or not.
>
> -To be short, we are the first the incorporate the representation disentanglement to FL system. The schema of disentangling the domain-invariant features proposed in our paper is novel and unique.
>
> -Generally, the discrepancy ${d}_{\mathcal{H}\Delta\mathcal{H}}({\mathcal{D}}_{S},{\mathcal{D}}_T)$ would be smaller when defined on the disentangled feature space. Thus, it's not accurate to claim "Representation disentanglement has no relation with the derived theory". Deep neural networks are known to extract features in which multiple hidden factors are highly entangled [5], this is especially true in the unsupervised federated domain adaptation scenario. Representation disentanglement has been proposed to explore the real-world explanatory factors. In our case, we leverage representation disentanglement to distill the domain-invariant knowledge and dispel the domain-specific information.

---

### Public Comment · ~Stone_Jamess1 · 2019-10-14
**Can you explain why use the feature disentanglement**

I have read this paper and know the authors would like to apply domain adaptation in a federated manner.
The idea is quite straightforward that multiple users will adapt their knowledge to one target domain.

I have the following questions.
1.  I am confused by the right part of Figure 1.  There are two CI in the figure, and what is the difference?
2. What is the output from G_i and G_t. In Equation (6), It says domain specific information is fed to DI but n Eq.(4), it says it is a local feature extractor. I am confused by the descriptions.
3 What is the benefit of using feature disentanglement, and how could feature disentanglement be used to improve the performance. Can you explain why you use feature disentanglement.


Thanks very much for your help.

---

> ### Author Response · Authors · 2019-10-15
> **Feature disentanglement facilitates the knowledge transfer by distilling out the key information**
>
> Thank you for your comment!
>
> In this paper, we propose a new scheme called Federated Adversarial Domain Adaptation (FADA) to address the domain shift in the distributed learning system, aiming to tackle the domain shift in the federated learning system by adversarial training. Federated learning improves data privacy and efficiency in machine learning performed over networks of distributed devices. However, existing federated work mainly focuses on data communication, data encryption, and training strategy for federated systems. To the best of our knowledge,  we are the first to propose a solid deep learning framework to tackle the domain shift between different nodes on the federated system.
>
> To answer your question:
> 1).  In Figure 1, the above CI is used to perform the recognition task (i.e. in image recognition task, it predicts the correct labels for the images) and it's trained with task loss ONLY! The below CI is utilized to enhance the disentanglement and it will be trained with cross-entropy loss and adversarial loss (in this paper we use entropy loss) iteratively.
>
> 2).  The output for G_i and G_t are the features for i-th source domain and target domain, respectively.
> In Equation 4, G_t and G_t extract features for the input data x^{s_i} and x^t.
>
> 3)  By disentangling the features, we are aiming to separate the deep features to domain-invariant features and domain-specific features. For example, if we forward an image (foreground: car || background: buildings, pedestrians || label: car) to a deep model, the network will encode the information from the foreground car and background buildings and pedestrians simultaneously. However, in the transfer learning task (car images in daylight -> car images at night ), only the information from the car is beneficial for the transfer task.
>
> We define the information which benefits the transfer task as domain-invariant features and the information which hampers or does not contribute to the transfer task as domain-specific features.
>
> Feature disentanglement can facilitate the knowledge transfer as it distills the key information for the transfer task.

---

> > ### Public Comment · ~Stone_Jamess1 · 2019-10-16
> > **Thanks for your reply**
> >
> > Thanks for your reply and I am still confused as the following.
> >
> > G_i includes both the domain-invariant and domain specific information. To enable the domain adaptation, the goal is to make DI cannot tell G_i and G_t.
> > If G_i includes both information, how could the Feature disentanglement facilitate the knowledge transfer. I am confused about it. In the experiment, I have seen that the experimental part proves that using feature disentanglement  can achieve the best performance.
> >
> > I just don't understand why since it lacks good theoretical explanation about it. The thing makes me confused is that even though the features from G_i are disentangled, but you still use both features and how could the DI tell if the feature are disentangled or not?
> >
> > Thanks again for your reply.

---

> > > ### Author Response · Authors · 2019-10-18
> > > **The benefit of feature disentanglement is not achieved by DI**
> > >
> > > Thank you for your reply.
> > >
> > > The feature disentanglement enables G_i to extract the domain-invariant features and dispelling the domain-specific features. The ultimate goal of employing feature disentanglement is to render a better G_i, and essentially, better G_t, since the gradient of G_t is the aggregation of gradient generated by each G_i. Note the task classifier is trained only on the domain-invariant features.
> > >
> > > The effect of feature disentanglement is not achieved by DI. DI is utilized to classify the features, i.e. to determine whether a feature vector is sampled from the source distribution or target distribution. By confusing the DI, we are aiming to align the source features with the target features.
> > >
> > > Thanks again for your reply!

---

### Decision · Program_Chairs · 2019-12-19

**Decision:**

Accept (Poster)

**Comment:**

This paper studies an interesting new problem, federated domain adaptation, and proposes an approach based on dynamic attention, federated adversarial alignment, and representation disentanglement.

Reviewers generally agree that the paper contributes a novel approach to an interesting problem with theoretical guarantees and empirical justification. While many professional concerns were raised by the reviewers, the authors managed to perform an effective rebuttal with a major revision, which addressed the concerns convincingly. AC believes that the updated version is acceptable.

Hence I recommend acceptance.